# Biomarker-Based Models for Preoperative Assessment of Adnexal Mass: A Multicenter Validation Study

**DOI:** 10.3390/cancers14071780

**Published:** 2022-03-31

**Authors:** Rafał Watrowski, Eva Obermayr, Christine Wallisch, Stefanie Aust, Nicole Concin, Elena Ioana Braicu, Toon Van Gorp, Annette Hasenburg, Jalid Sehouli, Ignace Vergote, Robert Zeillinger

**Affiliations:** 1Faculty of Medicine, University of Freiburg, 79106 Freiburg, Germany; rafal.watrowski@gmx.at; 2Molecular Oncology Group, Department of Obstetrics and Gynecology, Comprehensive Cancer Center-Gynecologic Cancer Unit, Medical University of Vienna, 1090 Vienna, Austria; eva.obermayr@muv.ac.at (E.O.); stefanie.aust@meduniwien.ac.at (S.A.); 3Section for Clinical Biometrics, Center for Medical Statistics, Informatics and Intelligent Systems, Medical University of Vienna, 1090 Vienna, Austria; christine.wallisch@meduniwien.ac.at; 4Department of Obstetrics and Gynecology, Innsbruck Medical University, 6020 Innsbruck, Austria; nicole.concin@i-med.ac.at; 5Department of Gynecology, European Competence Center for Ovarian Cancer, Campus Virchow Klinikum, Charité, Universitätsmedizin Berlin, 13353 Berlin, Germany; ioana@braicu.de (E.I.B.); jalid.sehouli@charite.de (J.S.); 6Division of Gynecological Oncology, Department of Obstetrics and Gynecology, Leuven Cancer Institute, University Hospitals Leuven, Katholieke Universiteit Leuven, 3000 Leuven, Belgium; toon.vangorp@uzleuven.be (T.V.G.); ignace.vergote@uzleuven.be (I.V.); 7Department of Obstetrics and Gynecology, Medical Center, University of Freiburg, 79106 Freiburg, Germany; annette.hasenburg@unimedizin-mainz.de; 8Department of Obstetrics and Gynecology, University Medical Center, 55131 Mainz, Germany

**Keywords:** ovarian cancer, adnexal mass, tumor marker, predictive models, diagnostic accuracy, multiplex, Luminex, adnexal mass, CA125, Copenhagen Index

## Abstract

**Simple Summary:**

Ovarian cancer (OC) is the deadliest genital tumor in women. In this multicenter study, we developed three new diagnostic models based on serum proteins and patient age. All models were then validated using data from centers other than those used for the model development. We also compared the performance of these models with common singular markers (CA125, HE4) and algorithms (ROMA-50 and the Copenhagen Index). We used modern technology (Luminex^®^, Austin, TX, USA) that enables the simultaneous determination of several biomarkers in a small amount of blood. A combination of patient age with four to six markers performed best: CA125, osteopontin, prolactin, macrophage migration inhibitory factor, and, eventually, HE4 and leptin. Our models were better than the ROMA-50 index but did not outperform the Copenhagen Index. In postmenopausal patients, all the newly developed models performed excellently. Unfortunately, none of the models tested improved the diagnosis in premenopausal patients and those missed due to normal CA125 levels.

**Abstract:**

Ovarian cancer (OC) is the most lethal genital malignancy in women. We aimed to develop and validate new proteomic-based models for non-invasive diagnosis of OC. We also compared them to the modified Risk of Ovarian Malignancy Algorithm (ROMA-50), the Copenhagen Index (CPH-I) and our earlier Proteomic Model 2017. Biomarkers were assessed using bead-based multiplex technology (Luminex^®^) in 356 women (250 with malignant and 106 with benign ovarian tumors) from five European centers. The training cohort included 279 women from three centers, and the validation cohort 77 women from two other centers. Of six previously studied serum proteins (CA125, HE4, osteopontin [OPN], prolactin, leptin, and macrophage migration inhibitory factor [MIF]), four contributed significantly to the Proteomic Model 2021 (CA125, OPN, prolactin, MIF), while leptin and HE4 were omitted by the algorithm. The Proteomic Model 2021 revealed a c-index of 0.98 (95% CI 0.96, 0.99) in the training cohort; however, in the validation cohort it only achieved a c-index of 0.82 (95% CI 0.72, 0.91). Adding patient age to the Proteomic Model 2021 constituted the Combined Model 2021, with a c-index of 0.99 (95% CI 0.97, 1) in the training cohort and a c-index of 0.86 (95% CI 0.78, 0.95) in the validation cohort. The Full Combined Model 2021 (all six proteins with age) yielded a c-index of 0.98 (95% CI 0.97, 0.99) in the training cohort and a c-index of 0.89 (95% CI 0.81, 0.97) in the validation cohort. The validation of our previous Proteomic Model 2017, as well as the ROMA-50 and CPH-I revealed a c-index of 0.9 (95% CI 0.82, 0.97), 0.54 (95% CI 0.38, 0.69) and 0.92 (95% CI 0.85, 0.98), respectively. In postmenopausal women, the three newly developed models all achieved a specificity of 1.00, a positive predictive value (PPV) of 1.00, and a sensitivity of >0.9. Performance in women under 50 years of age (c-index below 0.6) or with normal CA125 (c-index close to 0.5) was poor. CA125 and OPN had the best discriminating power as single markers. In summary, the CPH-I, the two combined 2021 Models, and the Proteomic Model 2017 showed satisfactory diagnostic accuracies, with no clear superiority of either model. Notably, although combining values of only four proteins with age, the Combined Model 2021 performed comparably to the Full Combined Model 2021. The models confirmed their exceptional diagnostic performance in women aged ≥50. All models outperformed the ROMA-50.

## 1. Introduction

Ovarian cancer (OC) accounts for 3.4% and 4.7% of new cancers and cancer-related deaths in women worldwide, with 313,959 new cases and 207,252 cancer-related deaths in 2020, respectively [1]. Globally, OC incidence, related deaths and disability-adjusted life-years doubled between 1990 and 2020 [1,2,3]. The overall mortality attributed to OC exceeds that of any other gynecological cancer [1,2,3]. This deadliest amongst all genital cancers will develop in 1.1–1.5% of women during their lifespan [1,2,3]. The high mortality rates result from usually advanced stages at diagnosis (70% of OC are diagnosed in FIGO stage III/IV), the high biological aggressiveness of the most common histological subtype (high-grade serous OC (HGSOC) accounting for 75% of all ovarian malignancies) and the lack of effective screening strategies [2,3,4,5]. Early diagnosis of OC is critical because the 5-year recurrence rates of OC in FIGO stage I are only 20% and the 10-year overall survival rates between 62% and 78% [6]. In contrast, the 5-year overall survival rates shrink to 35–50% as soon as OC disrupts the boundaries of the reproductive organs (>FIGO IIa) [7].

The mainstay of OC treatment is cytoreductive surgery combined with chemotherapy and targeted therapies. The prognosis of OC is strongly influenced by the quality of the primary therapy, especially the completeness of the surgical cytoreduction, which in turn correlates with the specialization of the surgeon [8]. Therefore, a reliable preoperative diagnosis of an adnexal mass should minimize the likelihood of overlooking malignancy, prompt the referral to a specialized center, and—otherwise—minimize the rate of unnecessary interventions. However, OC symptoms (bloating, abdominal pain, altered miction frequency) are unspecific and associated with already progressed disease. If an adnexal tumor is suspected, the therapeutic decision is mainly based on transvaginal ultrasound (TVS), possibly supplemented by other imaging methods, and the evaluation of serum biomarkers. Unfortunately, TVS is not ubiquitously accessible at expert level and 25% of the tumors lack features clearly indicating benignity or malignancy [9].

Biomarkers are “compounds present in or produced by cancer cells or by other cells of the organism in response to and in correlation with the tumor” [10]. Circulating biomarkers can be found in blood, ascites, or other body fluids. The requirements for biomarkers depend on whether they are used for diagnostic purposes (screening of asymptomatic women, differential diagnosis of adnexal mass, post-treatment surveillance), estimation of the prognosis (clinical outcomes regardless of treatment) or prediction of response to therapeutic interventions. According to their biological role, tumor markers include: circulating cancer cells or tumor cell components (e.g., cell-free DNA, miRNA, extracellular vesicles); surface antigens, adhesion molecules or proteases (e.g., cancer antigen 125 [CA125], human epididymis protein 4 [HE4], osteopontin [OPN], mesothelin); hormones, growth factors and related molecules (e.g., prolactin [PRL], leptin, insulin-like growth factors [IGF] and their binding proteins); cytokines (e.g., macrophage migration inhibitory factor [MIF], interleukins [IL] and their soluble receptors); autoantibodies (anti-TP53, anti-IL-8) or inflammatory markers (e.g., C-reactive protein, fibrinogen, blood platelets, neutrophil-to-lymphocyte ratio, platelet-to-lymphocyte ratio) [10,11,12,13].

Since the first assay became available in 1983 [14], CA125 remains the most utilized marker for the diagnosis and monitoring of OC. Circulating CA125 levels increase with tumor burden, but—conversely—they do not exceed the usual cutoff of 35 U/mL at 50% of early OC [15]. The sensitivity of CA125 has been reported at 0.78–0.86, the specificity at 0.57–0.82, the positive predictive value (PPV) at 0.50–0.68 and the negative predictive value (NPV) at 0.9–0.95 [11,12,16,17,18,19]. Amongst a further 200 investigated biomarkers, the cancer antigens CA 72.4, CA 15.3, CA 19.9 and HE4 have been considered the most informative for the evaluation of adnexal mass [4,13,20]. HE 4 received particular attention as initial studies indicated it had better specificity (0.78–0.94) with sensitivity (0.75–0.86), PPV (0.62–0.63) and NPV (0.87–0.96) being comparable to those of CA125 [17,18]. In 2009, a Risk of Ovarian Malignancy Algorithm (ROMA) including CA125, HE4 and menopausal status was proposed [21]. A further combination of CA125 and HE4 with three biomarkers dysregulated in OC (transferrin, apolipoprotein A1, follicle stimulating hormone) was launched as the OVERA test, a spin-off of the former OVA-1 test [22]. In their registration trials, both the ROMA and OVERA predicted pelvic malignancy with 0.91–0.94 sensitivity, 0.69–0.74 specificity and 0.97–0.99 NPV [5]. ROMA, OVA1 and OVERA have been cleared by the US Food and Drug Administration (FDA) for diagnosing suspicious adnexal mass, but not for screening [19,23]. A promising modification of ROMA is the Copenhagen Index (CPH-I), calculated from CA125, HE4 and patient age, which seems to be equal or superior to ROMA, and does not require knowledge of the patient’s menopausal status [24,25,26,27].

In 2008, based on the protein selection of Mor et al. [28], Visitin et al. [29] proposed a diagnostic panel consisting of CA125, OPN, PRL, leptin, IGF-2 and MIF. The reported 95.3% sensitivity and 99.4% specificity prompted the introduction of this biomarker combination as a commercially available blood test (OvaSure^TM^). However, the scientific community [30,31,32] and the FDA [33] raised serious concerns about the quality of the validation study, which finally led to the correction of the overestimated results and the withdrawal of OvaSure^TM^ from the market [29,33]. Despite the shortcomings of the original study, our working group independently utilized the six-marker combination. In 2013, we developed a multi-omics model combining the six-protein panel with two gene signatures achieving a sensitivity of 97.8%, a specificity of 99.6%, and an AUC of 0.998 [34]. In a study including patients with BRCA1 mutation, we observed an excellent performance of the six-protein panel (modified by replacing IGF-2 with HE4) both in OC patients with BRCA1 mutation (AUC 0.98) and wildtype (AUC 0.99). Of particular interest was that the test enabled differentiation between healthy BRCA1 mutation carriers and healthy wildtype women (with 0.88 sensitivity, 0.81 specificity and AUC 0.9) [35]. Lastly, we re-estimated the six-protein panel and calculated the Proteomic Model 2017, which consisted of CA125, HE4, PRL, OPN and leptin, and reached an AUC of 0.96 in the training cohort [36]. To date, none of these models have been independently validated.

The aim of the present study was to develop and validate new predictive models for non-invasive differentiation between benign and malignant adnexal mass. For comparison, we also validated the modified ROMA (ROMA-50), the CPH-I and the Proteomic Model 2017. As secondary aims, we evaluated the predictive ability of the new models in patients with normal vs. elevated CA125 concentrations (cutoff 35 U/mL), and in patients under and above 50 years of age. 

## 2. Materials and Methods

### 2.1. Patients and Samples

This was a multicenter study at five European gynecological-oncological centers. The study was conducted in accordance with the Declaration of Helsinki and approved by the Institutional Review Boards (Ethics Committees) at the University Hospitals of Vienna (EK366/2003), Innsbruck (UN5067), Leuven (S54404), Berlin (207/2003) and Freiburg (59/13). All patients gave their written informed consent. Patients with pre-existing malignancy were excluded from the study.

The clinicopathological classification and staging were carried out according to the classifications of the WHO (2014) [37] and FIGO (2013) [38]. Patient-specific and clinical-pathological data were stored anonymously in a database and handled according to good scientific practice.

Blood samples were obtained in 2018 and 2021 from patients with a suspicious ovarian mass 24–48 h prior to primary surgery. The blood was collected in EDTA tubes, and the cellular components were removed by centrifugation. Biomaterials were processed and stored until analysis according to standard operating procedures by the MedUni Wien Biobank, a central facility included in a certified quality management system [39]. In some of the samples, the blood plasma was retrieved after a two-layer density gradient centrifugation and stored in aliquots at −80 °C until analysis.

### 2.2. Laboratory Analyses

All analyses were performed in the laboratory of the coordinating center (Molecular Oncology Group, Medical University of Vienna, Vienna, Austria) employing the multiplex bead-based Luminex^®^/xMAP^®^ technology. The 96-well plate assay panel (MILLIPLEX MAP Human Circulating Cancer Biomarker Magnetic Bead Panel 1; EMD Millipore, Billerica, MA, USA) was used to quantify the six proteins (CA125, HE4, OPN, PRL, leptin, and MIF) according to the manufacturer’s instructions. After completion of all incubation and washing steps the 96-well plate was read with the Bio-Plex™ 200 array reader (Bio-Rad Laboratories, Hercules, CA, USA). Raw data were analyzed with the Bio-plex Manager Software vs 6.1 (Bio-Rad). The Median Fluorescent Intensity (MFI) was assessed using a 5-parameter logistic curve-fitting method to calculate the concentration of all analytes.

### 2.3. Statistical Analysis

Categorical variables are presented as frequencies and proportions, while continuous variables as means with standard deviation (SD). The predictive ability of individual markers and models was evaluated by the Receiver Operating Characteristic (ROC) curve, the area under the ROC curve (also called c-index) and corresponding indicators of diagnostic accuracy (overall accuracy, sensitivity, specificity, PPV, NPV). The optimal classification cut-point was defined individually based on the maximum Youden’s index, which is sensitivity + specificity − 1. A c-index close to 0.5 means that the model is not better than randomly assigning outputs, whereas values of ≥0.9 confirm excellent discrimination. 

For the development of the Proteomic, the Combined and the Full Combined Model 2021 we used a logistic regression model to estimate the risk of OC and selected relevant predictors by applying elastic-net regularization (with α = 0.5). With elastic-net regularization, some coefficients were shrunken to zero by penalizing coefficients with a penalty parameter, thus, the corresponding predictors were not selected. The penalty parameter was chosen by leave-one-out cross-validation using the deviance as optimization criterion. Regularization attempts to correct for overfitting and could lead to better performance in the validation cohort. Prior to model fitting, we added 0.5 to the values of the six proteins to avoid infinite values after a base-2 logarithmic transformation. 

All six proteins were included in the pool of potential predictors for the Proteomic Model 2021, which is based on the elastic-net approach. For the Combined Model 2021, we included age in addition. For comparison, we fitted the Full Combined Model 2021, which also included age and all six protein markers, but ridge regularization was used instead of elastic-net. This approach shrinks coefficients towards zero but does not set them exactly to zero such that all variables remain included in the model. The penalty parameter of the Full Combined Model 2021 was again chosen by leave-one-out cross-validation using the deviance as optimization criterion. 

For comparison to other models, we validated the modified ROMA, the Copenhagen Index (CPH-I) and our earlier Proteomic Model 2017. 

The ROMA Index was calculated as a combination of CA-125, HE4 and menopausal status. Because exact data on menopausal status were not available, we defined the menopausal status as: age < 50 years for “pre-menopausal” and age ≥ 50 years “post-menopausal”. To indicate this modification, we named the algorithm ROMA-50. The ROMA-50 index was calculated according to the following formula [21]:Pre-menopausal: −12.0 + 2.38 × LN(HE4) + 0.0626 × LN(CA125)
Post-menopausal: −8.09 + 1.04 × LN(HE4) + 0.732 × LN(CA125)

In some patients, marker values of CA-125 and HE4 were zero. To be able to calculate the ROMA-50 Index, we added 0.1 to all values.

The CPH-I was calculated as a combination of CA-125, HE4 and age (as continuous variable) using the following formula [24]:CPH-I = −14.0647 + 1.0649 × log_2_(HE4) + 0.6050 × log_2_(CA125) + 0.2672 × Age/10

The risk of OC based on the Proteomic Model 2017 was calculated as a combination of CA-125, HE4, OPN and PRL according to the following formula [36]: (1 + exp (−(−27.6312 + log_2_(CA125 + 0.5) × 0.6749 + log_2_(OPN + 0.5) × 1.9572 + log_2_(HE4 + 0.5) × 0.2234 + log_2_(leptin + 0.5) × −0.1320 + log_2_(PRL + 0.5) × −0.2910))^−1^

The statistical analyses were performed using R version 4.0.4 (R Foundation for Statistical Computing, Vienna, Austria). For all statistical tests, the level of significance was set at *p* = 0.05.

## 3. Results

### 3.1. Patient Characteristics

The dataset comprised 356 women with suspicion of ovarian carcinoma. 219 women were included in 2018 and 137 were added in 2021. In total, 106 benign tumors, 12 borderline tumors, and 238 malignant tumors were diagnosed. We developed models based on the data of the centers in Freiburg, Berlin, and Vienna (*n* = 279). The data of the centers in Innsbruck and Leuven constituted the validation cohort (*n* = 77). The overall prevalence of OC in the training and validation cohorts were 73% (203 of 279) and 61% (47 of 77), respectively. Borderline cases (*n* = 10 in the training cohort and *n* = 2 in the validation cohort) were included in the group of malignant tumors. Most patients of both cohorts were diagnosed with advanced OC stages (FIGO IIb-FIGO IV), at 84% in the training cohort and at 95% in the validation sample. The mean age of patients in the training and validation cohort was 55.2 ± 15.6 years and 56.3 ± 15.6 years, respectively. The descriptive statistics of the training and validation datasets is summarized in Table 1. The distribution of histological subtypes within each cohort is provided in Table A1 (Appendix A). Outcome statistics broken down by participating center are presented in Table A2.

### 3.2. Performance of Individual Markers

The visualization of the protein values in the validation cohort is presented in Figure A1. The pairwise correlation structure of all predictors is shown in Figure A2. Only pairs PRL/OPN and CA125/HE4 achieved correlation coefficients above 0.5. Age correlated weakly (at about 0.3) with CA125, HE4 and MIF.

The diagnostic performance of singular predictors is shown in Figure 1 and Table 2.

According to their AUC value, protein markers ranked individually as follows: CA125 (0.87) > OPN (0.81) > HE4 (0.8) > leptin (0.7) > MIF (0.55) > PRL (0.51). Remarkable was the second-best performance of OPN, which was slightly better than of HE4, whereas the discriminative ability of PRL or MIF appeared negligible. 

Age, the only non-proteomic predictor, demonstrated—at the threshold of 51.5 years—a sensitivity of 0.85, a specificity of 0.67 and a c-index of 0.76 (95% CI 0.64–0.88). Thus, the predictive ability of age was better in comparison to leptin, MIF or PRL.

As shown in Table 2, the sensitivity of OPN (0.89) at the optimal cutoff was the highest of all markers and models examined. In contrast, the specificity of OPN (0.7) at the calculated cut-point was among the poorest of any protein marker along with leptin (0.7). Notably, PRL’s worst overall accuracy (0.51) was the product of its critically poor sensitivity (0.21) and NPV (0.44), despite its outstanding specificity (0.97) and high PPV (0.91).

### 3.3. New Predictive Models

Regression coefficients of all models are given in Table A3 (Appendix A). The discriminative ability of the new models in the validation cohort is shown in Table 2 and Figure 2.

#### 3.3.1. Proteomic Model 2021: CA125, OPN, MIF, and PRL

The model was built on the logarithmized data (base2, first adding 0.5 to the data).

The Proteomic Index 2021 (resulting from the Proteomic Model 2021) is:−4.7705 + 0.3699 × log_2_(MIF + 0.5) + 0 × log_2_(Leptin + 0.5) + 0.6134 × log_2_(CA125 + 0.5) + 0.0003 × log_2_(PRL + 0.5) + 0.0159 × log_2_(OPN + 0.5) + 0 × log_2_(HE4 + 0.5)

The predicted probability for ovarian malignancy based on the Proteomic Model 2021 is calculated as:$$\frac{1}{1+\exp\left(1-\mathrm{Proteomic}\;\mathrm{Index}\;2021\right)}$$

In this model Leptin and HE4 were not selected by the algorithm (coefficients for these two proteins were 0). The Proteomic Model 2021 revealed a c-index of 0.98 (95% CI 0.96, 0.99) in the training cohort. As shown in Table 2, in the validation cohort it could only achieve a c-index of 0.82 (95% CI 0.72–0.91). Remarkably, the Proteomic Model 2021 demonstrated the best specificity (0.97) and PPV (0.97), but the worst sensitivity (0.64) and second-worst NPV (0.63) of all models examined.

#### 3.3.2. Combined Model 2021: Selection of Proteins + Age

For the Combined Model 2021, age was added as a potential predictor and the resulting formula for the Combined Index 2021 is:−8.3526 + 0.0621 × log_2_(MIF + 0.5) + 0.3419 × log_2_(Leptin + 0.5) + 0 × log_2_(CA125 + 0.5) + 0.5455 × log_2_(PRL + 0.5) + 0.0259 × log_2_(OPN + 0.5) + 0.0559 × log_2_(HE4 + 0.5)

The predicted probability for ovarian malignancy based on the Combined Model 2021 is calculated as:$$\frac{1}{1+\exp\left(1-\mathrm{Combined}\;\mathrm{Index}\;2021\right)}$$

The Combined Model 2021 achieved a c-index of 0.99 (95% CI 0.97, 1) in the training cohort (not optimism-corrected). In the validation cohort, it obtained a c-index of 0.86 (95% CI 0.78–0.95). Remarkably, although utilizing values of only four proteins, the sensitivity (0.83), specificity (0.90), PPV (0.93), NPV (0.77) and overall accuracy (0.86) of the Combined Model 2021 were only slightly lower as compared to the Full Combined Model 2021, which is presented below.

#### 3.3.3. Full Combined Model 2021: All Proteomics + Age

In contrast to the Combined Model 2021, the Full Combined Model 2021 included all protein markers and age, and the Full Combined Index 2021 is:−6.7962 + 0.0465 × log_2_(MIF + 0.5) + 0.2385 × log_2_(Leptin + 0.5) + −0.034 × log_2_(CA125 + 0.5) + 0.3346 × log_2_(PRL + 0.5) + 0.0428 × log_2_(OPN + 0.5) + 0.0885 × log_2_(HE4 + 0.5)

The predicted probability for ovarian malignancy based on the Full Combined Model 2021 is calculated as:$$\frac{1}{1+\exp\left(1-\mathrm{Full}\;\mathrm{Combined}\;\mathrm{Index}\;2021\right)}$$

The Full Combined Model 2021 achieved a c-index of 0.98 (95% CI 0.97, 0.99) in the training cohort whereas it obtained a c-index of 0.89 (95% CI 0.81–0.97) in the validation cohort. The sensitivity (0.83), specificity (0.93), PPV (0.95), NPV (0.78) and overall accuracy (0.87) of the Full Combined Model 2021 were the best amongst all investigated models. 

### 3.4. Comparison to Other Predictive Models

For comparison, we have validated three already existing diagnostic models, whose discriminative ability is shown in Figure 2 and Table 2.

#### 3.4.1. Proteomic Model 2017

The Proteomic Model 2017 included five protein markers: CA125, PRL, leptin, OPN and HE4. In the validation cohort, the model obtained a c-index of 0.9 (95% CI 0.82, 0.97) This was lower than the c-index of 0.96 in the development cohort (reported in Ref. [36]). 

#### 3.4.2. ROMA-50

In the validation cohort, the modified ROMA Index (ROMA-50) achieved a c-index of only 0.54 (95% CI 0.38–0.69). This was the lowest c-index of all models studied. ROMA-50 also showed the poorest specificity (0.67), PPV (0.78), NPV (0.62) and overall accuracy (0.71) among all models.

#### 3.4.3. CPH-I

In the validation cohort, the CPH-I achieved a c-index of 0.92 (95% CI 0.85–0.98). The sensitivity (0.83), specificity (0.90), PPV (0.93), NPV (0.77) and overall accuracy (0.86) of the CPH-I were almost identical with those of the Combined Model 2021 (see Table 2).

The inclusion of age as continuous variable to any of the predictive models significantly improved their diagnostic performance. Sensitivity (0.83) and PPV (0.93) of all models including age (Combined Model 2021, Full Combined Model 2021, CPH-I) were identical. There were also no differences between specificity (0.90) and NPV (0.77) of the Combined Model 2021 and CPH-I, as well as between specificity (0.93) and NPV (0.78) obtained by the Full Combined Model 2021 and the Proteomic Model 2017. 

Overall, the CPH-I, both Combined Models of 2021, and the Proteomic Model 2017 showed satisfactory diagnostic accuracies, without clear superiority of only one model. All models outperformed the ROMA-50. CA-125 and OPN showed the best discriminatory ability as single markers.

### 3.5. Subanalyses

#### 3.5.1. Age < 50 vs. Age ≥ 50 Years

In the validation cohort, 27 patients (35%) were aged below 50 years, whereas 50 (65%) were older than 50 years (roughly to consider as “postmenopausal”). As shown in Table 3 and Figure 3, the discriminatory ability of the Proteomic Model 2021, the Combined Model 2021 and the Full Combined Model was excellent for patients aged ≥ 50 years, showing a sensitivity of 0.9, 0.92 and 0.95, respectively, with a specificity of 1.00 and PPV of 1.00 in every case. In contrast, all three newly developed models performed like a coin flip for patients younger than 50 years. Confidence intervals were too wide to identify a preferred model for the diagnosis of malignant tumors in women aged 50 or older. 

#### 3.5.2. Normal CA-125 vs. Elevated CA-125

In the validation cohort, 38 patients (49% of the validation sample) had normal CA125 levels, while 39 patients (51%) had elevated CA125 values. Among all CA125 negative patients, 11 cases were false-negative (11 of 47 malignant cases), representing a 23% false negative CA125 rate. As shown in Table 4 and Figure 3, in patients with normal CA-125 levels, all models were poorly able to discriminate between benign and malignant tumors. Only the Full Combined Model 2021 appeared to perform better (AUC 0.67) than both other models, but again the confidence intervals were too wide to draw reliable conclusions for the preference of one model.

## 4. Discussion

Detection of asymptomatic disease at initial stages could reduce OC mortality by 10–30% [5]. This goal remains unattainable, since no available screening strategy guarantees the necessary specificity of >99.6% with a sensitivity of at least 75% [4,5]. Nonetheless, treatment outcomes can be improved by a prompt referral to a highly specialized gynecological surgeon [8]. In this situation, there is an unmet need for reliable biomarkers. Given the biological heterogeneity of OC, a single biomarker is unlikely to detect all cases of OC. For instance, CA125 is overexpressed in 85% of the serous and 65% of the endometrioid, but only in 40% of the clear cell and 12% of the mucinous subtypes of OC [40]. Multi-marker panels offer the opportunity to reflect the heterogeneity of cancer cells and to improve the diagnostic triage. The Luminex^®^/xMAP^®^ technology is a bead-based immunoassay that allows for multiplex detection of up to 100 analytes simultaneously [41]. The commercially available bead panel—applied in [29,42,43,44,45] and our research [34,35,36]—offers a wide spectrum of pre-defined biomarkers. Selection and validation of the best performing marker panels is essential for their reasonable and cost-effective use.

In 2008 Visintin et al. [29] described the Luminex^®^ based panel of CA125, IGF-2, MIF, PRL, OPN and leptin as “the first blood biomarker test with a sensitivity of 95.3% and a specificity of 99.4% for the detection of ovarian cancer”. Based on this study, this six-protein signature was promptly launched under the trade name OvaSure^TM^ as the long-awaited screening test for OC. Unfortunately, the study earned criticism, mainly because it would “violate fundamental principles of statistical analysis” [32] by combining the data from the training and validation cohorts to improve the results obtained in the validation cohort alone. The further concern was that the claimed PPV of 99.3% would presume a 50% point prevalence of OC, whereas the expected prevalence in the screening population was 0.04% [30]. In consequence, the re-estimated results (84% sensitivity, 95% specificity, 6.5% PPV) were published and OvaSure^TM^ was withdrawn from the market [29,30,32]. 

The six-protein signature was sparsely investigated in the following years. In the study of Vrzalova et al. [42], using the same multiplex panel, only CA125, OPN and IGF-2 differed significantly between patients with malignant and benign tumors, while serum levels of MIF, PRL and leptin did not. Therefore, the authors recommended including HE4 in the multiplex panel. Gschwantler-Kaulich et al. [35] noticed an excellent performance of such a modified panel in women carrying the BRCA1 mutation. Lu et al. [43] analyzed 14 different markers using the Luminex^®^ technology and identified seven markers that were differentially expressed in the presence of benign and malignant ovarian tumors. Remarkably, they included all six markers described in [28,29] beside of p53-autoantobodies (AAbs). Of note, the diagnostic performance of the CA125/p53-AAbs combination was equivalent to that of the six-marker signature (AUC 0.969 and 0.972, respectively). Applying a similar multiplex approach, Guo et al. [44] assessed concentrations of 10 protein antigens and 12 autoantibody biomarkers, finally obtaining a four-biomarker panel (CA125, OPN, MIF, and anti-IL-8-AAbs). The validation of this four-marker panel revealed an AUC 0.974 for early-stage OC, and AUC 1.00 for late-stage carcinoma. Horala et al. [45] examined 18 markers (16 from the Luminex^®^ panel and conventionally measured CA125 and HE4) and finally recommended a three-marker panel consisting of OPN, CA125 and HE4 reaching a c-index of 0.958. Lastly, based on the modified six-protein panel we developed a five-protein signature—Proteomic Model 2017 (CA125, HE4, OPN, leptin and PRL) that revealed a c-index of 0.96 [36]. In 2021, Walker et al. [46] re-evaluated the six-protein panel previously reported by their group [28,29]. Leptin and IGF-II, having the lowest sensitivity and specificity, were removed from the model. CA125, MIF, OPN, and PRL were quantified using an automated multiplex platform and evaluated using four classification techniques. Their four-protein model yielded a c-index of 0.935 [46]. These results are of particular interest because the protein selection is identical with that of our Proteomic Model 2021.

All the protein-only models pursued two ideas: a simultaneous measurement of multiple markers in a minimal amount of blood (which is enabled by multiplex technologies), and the discovery of biomarker patterns that allow for reliable, potentially remote diagnosis, regardless of TVS availability or knowledge of the patient’s symptoms and medical history. Here, we developed three and validated a total of six predictive models. Applying the same methodology as used to develop the Proteomic Model 2017 [36], we generated an actualized model which now included MIF, but not HE4 and leptin. The Proteomic Model 2021 was thus defined as a combination of CA125, OPN, PRL and MIF.

Age is an easily accessible and informative variable. Combining age with the four proteins yielded the Combined Model 2021. This improvement demonstrated near-perfect discriminative ability in the training cohort. In contrast to the variable-reducing approaches, in the Full Combined Model 2021 we extended the initial six-protein panel by adding age. We believe that age, necessary for patients’ identification and always available, should be routinely considered when developing diagnostic tools for OC. Of note, we noticed that the predictive ability of age was better compared to leptin, MIF or PRL. Not surprising that the performance of CA125, HE4 and age—blended to CPH-I—demonstrated the highest c-index among all investigated marker combinations. Here, substituting the dichotomous menopausal status (ROMA) for age (CPH-I) radically improved the performance of the two-marker combination CA125/HE4. 

While all new models in the training cohort performed excellently, the validation process revealed a more nuanced picture. The Proteomic Model 2017 (developed in a different, much smaller training cohort) performed non-significantly better than the Proteomic Model 2021, as the latter demonstrated the highest specificity but the lowest sensitivity among all validated models. These differentiated findings confirm that validation in independent cohorts is essential when developing diagnostic tools. The results in the validation cohort reflect the expected performance in a new cohort or population. It is not surprising that markers with lower diagnostic performance are more likely to be interchangeable between models. In addition, the performance of each model was calculated according to the determined optimal cutoff. It might be possible that the performance indicators would slightly change in case the predefined fix cutoffs had been used.

Nevertheless, we discussed pre-analytical bias as a possible explanation for the lower performance of the Proteomic Model 2021, as we found that mean MIF levels were unexpectedly similar between benign and malignant cases at one of the two validation centers (data not shown). It naturally would lower the diagnostic ability of MIF, which was part of the 2021 panel. We were able to rule out any laboratory error, so we finally accepted this anomaly as an expression of the natural diversity known for almost all markers.

Both the Combined Model 2021 and the Full Combined Model 2021 confirmed their good diagnostic performance in the validation cohort. Interestingly, sensitivity and PPV of all models considering age (Combined Model 2021, Full Combined Model 2021, CPH-I) were virtually identical. Overall, the new models performed similarly and very well. The combination of markers likely stabilized performance and elevated it to a high but limited level around 0.9, as neither model reached the AUC values of 0.98–0.99 observed in the training cohorts.

The concurrent validation of ROMA-50 and CPH-I produced the most polarized results within the whole validation cohort: ROMA-50 had a c-index of 0.54, while CPH-I had one of 0.92. The poor performance of ROMA-50 was below expectations and below the reported ranges for ROMA and its variants [19,23,27]. We think that the unsatisfactory performance of ROMA is due to the fact that the dichotomous variable menopause (which summarizes e.g., early postmenopausal and octogenarian women) is less informative compared to age, which reflects the continuous accumulation of cancerogenic defects and mutations together with non-oncologic comorbidities throughout the lifespan. Regarding CPH-I, the findings of our study, albeit beyond its primary objective, are a valuable contribution to the sparse literature on CPH-I. We have embedded our results into the existing literature body in Table A4 (Appendix A).

The most intriguing results were obtained in women over 50 years of age. All three newly developed models achieved a specificity and a PPV of 100%, and a sensitivity of over 90%. However, it should be noted that this excellent discriminative ability was obtained in a validation cohort with an OC rate of 61%, while in the general population aged 50 and older the estimated annual incidence of OC is approximately 0.04% [31]. Therefore, a validation study in a postmenopausal cohort with an expected OC rate < 1% is warranted to determine the suitability of the new models for screening purposes. 

Regarding singular markers, three markers (CA125, OPN, HE4) ranked moderately high (c-index above 0.8), and two other (MIF and PRL) poor (c-index around 0.5). 

The performance of CA125 was largely consistent with the literature data. In this study, the specificity of CA125 was comparable to HE4, along with a higher sensitivity. To date, despite limitations, no single marker has been able to outperform CA125 in the diagnosis and follow-up of ovarian mass. The pitfalls associated with CA125 are rooted in its biological role. Also known as mucin 16, CA125 is a transmembrane glycoprotein that is undetectable in normal ovarian epithelium but is highly expressed in OC cells and is relevant for tumor growth, adhesion, and metastasis [15]. In addition, CA125 is normally expressed in tissues derived from the coelomic epithelium (pleura, pericardium, and peritoneum) and in distant organs such as lung, breast, or conjunctiva [15,47]. Any process disrupting the epithelial lining of the peritoneum has the potential to increase CA125 levels, so its elevations are observed in the presence of endometriosis, adenomyosis, ovarian and uterine fibroma, pregnancy, menstruation, liver cirrhosis, abdominal tuberculosis, heart insufficiency and non-ovarian (e.g., uterine, pulmonary, or gastrointestinal) malignancies [15,16]. In general populations, 10% of abnormal CA125 results are attributable to OC, whereas further 12% to another carcinoma [16]. Beyond oncology, CA125 has been used for the monitoring of endometriosis or congestive heart failure [48,49].

The FDA-approved biomarker HE4 has proven to be more specific than CA125, is insensitive to endometriosis and thus particularly interesting for diagnosing premenopausal women [23]. HE4 is a secreted glycoprotein that is overexpressed by 100% of endometrioid, 93% of serous, 50% of clear cell, and 0% mucinous OC. It is present in the reproductive tract and respiratory epithelium, but not in the normal ovarian surface epithelium [50]. In malignancy, HE4’s function is related to adhesion, migration, and growth of cancer cells via multiple signaling pathways [51]. Most non-ovarian carcinomas lack HE4 expression [50]. In our study, HE4 provided a slightly better specificity than CA125, however, sensitivity was lower than that obtained for age, CA125, OPN and leptin. It was unexpected that the mathematical algorithm deselected HE4 during the model building process, as it added no relevant information over the four-protein panel.

At the optimal cut-off value, OPN demonstrated the best sensitivity among all single markers in our dataset. Due to only moderate specificity, its overall performance was behind CA125 but ahead of HE4. Our results were similar to [45], a study applying the same multiplex technology. OPN is a multifunctional phosphoglycoprotein that is expressed by normal osteoclasts, osteoblasts, neurons, epithelial cells of the breast kidney and skin, vascular smooth musculature, as well as various (T, B, NK) lymphocyte populations and macrophages [52]. It is involved in wound healing, angiogenesis, inflammation, immune response, and tumorigenesis [45,52,53]. OPN is elevated in peripheral blood of human cancer patients, including OC [45,53,54,55], and correlates with clinical stage, histological grade, and lymph node involvement [52,53]. OPN contributes to cancer aggressiveness by promoting epithelial-mesenchymal transition (EMT) and metastasis, activation of cell proliferation, chemotherapy and radiation resistance, and distortion of immune function [52]. OPN has been recognized as a potential OC biomarker for 20 years [53] and demonstrated—in more than two dozen studies—a mean sensitivity around 0.7 and a specificity close to 0.9 [54,55]. Nevertheless, it has not been incorporated into established diagnostic algorithms.

The performance of PRL among other markers examined was unique as it demonstrated excellent specificity along with an unacceptably low sensitivity. However, the low sensitivity of this marker is not surprising. PRL is a circulating pituitary polypeptide hormone, measurable in healthy women without any ovarian pathology at levels partially overlapping the ranges obtained in this study. Although the main source of the circulating PRL are hypophyseal lactotrophic cells and its best recognized physiological role is the regulation of implantation, pregnancy, lactation and sexual response, PRL is also synthesized at several sites outside the pituitary, e.g., the endometrium, myometrium, decidua, immune cells, brain, breast, prostate, skin, and adipose tissue, where it is involved in neuromodulation, immunoregulation, angiogenesis, cell proliferation and differentiation [56]. In gynecologic cancers, PRL exhibits antiapoptotic and pro-tumoral properties by promoting tumor cell migration, invasion, metastasis, and chemoresistance via various signaling pathways and effector proteins [56].

In this study, MIF was the last marker included in the models. MIF is a multipotent cytokine expressed by various cell types and tissues, e.g., cells of the immune and nervous systems, epithelial, endothelial, smooth muscle, and mesenchymal cells [57]. MIF mediates host innate and adaptive immunity and regulates survival pathways, important for pathogen elimination during infectious disease. As an immunomodulator, MIF accelerates inflammation, promotes metastasis and cancer progression [57]. MIF is abundant in borderline and malignant ovarian tumors. Autocrine production of MIF by OC cells has been shown to stimulate other cytokines, chemokines, and angiogenic factors that may promote peritoneal invasion and neovascularization [58].

The overall performance of leptin was mediocre, which led to its exclusion from all 2021 models. Leptin is a pleiotropic hormone primarily produced by adipose tissue. Its best-known biological role is to signal satiety to the hypothalamus and subsequently reduce food intake and fat storage. Leptin was shown to promote cancer progression and metastasis by regulating the EMT, cell adhesion to the extracellular matrix (ECM), and proteolysis of the ECM compounds [59,60]. Although leptin’s expression is common in most OC cells and is associated with higher aggressiveness and poorer treatment outcome [59,60], circulating leptin levels are usually lower in presence of OC compared to benign tumors [28,29,61,62], as could also be observed in our setting.

The most disappointing results were obtained in patients younger than 50 years (roughly considered premenopausal) and in OC cases missed by false-negative CA125 measurements. In these challenging patient groups, the performance of the newly developed diagnostic models was statistically close to a coin toss. Other multivariate assays such as OVA1 (CA125, transferrin, transthyretin, beta-2-microglobulin, and apolipoprotein A-1) have been reported to detect 50% of OC cases in premenopausal women with CA125 levels below 67 U/mL, and up to 63% of early-stage OCs missed by CA125 [63]. Missing patients by negative CA125 values can either be due to a constitutional lack of CA125 expression in the tumor or to an only non-significant increase during the early development of a potentially curable cancer. Unfortunately, this major problem of the contemporary marker use could not be solved with our study. We hypothesize that diagnostic models created by adding biomarkers of tumor antigen type offer increased but still limited accuracy and, therefore, combining different categories of biomarkers, e.g., CA125 with inflammatory markers, antibodies and age would be the right way to discover maximally effective tools for OC diagnosis. Future studies could evaluate the utility of combining our diagnostic models with sonographic criteria [64] or body mass index [65].

In addition to several questions that could be answered or at least addressed by the present work, some questions remained unanswered due to the limitations of our study.

The first limitation was a low rate of early-stage carcinomas in the validation cohort. The performance of our models in patients with limited disease would be intriguing, since in the study of Guo et al. [44], a marker panel including three out of four markers incorporated into our “2021” models showed superior performance compared to CA125 alone, even in early-stage disease.

A further limitation is that our results are hardly generalizable to non-serous carcinomas, due to the dominance of HGSOC in our study population. However, HGSOC is the most common subtype, so the cohort was representative. On the other hand, mucinous, or clear cell OC are characterized by divergent marker expression patterns and therefore it would be interesting if the multi-marker panels match the expression patterns in these infrequent subtypes [40,50]. However, it may not be appropriate to extrapolate the results obtained in all CA125 false-negative cases to inherently CA125-negative subtypes, as the panels might improve the diagnosis ultimately.

The high proportion of OC in both the training and validation cohort should not necessarily be seen as a flaw. However, one should be aware that the indicators of diagnostic accuracy depend on prevalence of the index condition in the cohort. This general rule could be explicitly demonstrated for ROMA and its variants [66]. It should be noted that the majority of diagnostic performance studies was performed in collectives with high OC (e.g., 40%) prevalence [18]. Such cohorts are representative of e.g., referral centers. In contrast, it is to be expected that prevalence-dependent parameters like sensitivity, predictive values etc. can be different in the general population. Therefore, caution should be taken when models validated in cohorts with high OC prevalence are applied to populations without suspected ovarian tumor, e.g., for general screening purposes. Likewise, the performance indicators cannot be automatically adapted to populations with an increased OC prevalence due to e.g., BRCA mutation, since proteomic signatures may differ with regard to tumor biology.

To clarify the generalizability of the Proteomic Models to general screening populations or others, external validation studies need to be conducted since here only internal validation was performed with a retrospective study design and without sample size calculations.

Finally, direct comparisons of exact values obtained by Luminex^®^ and conventional methods should be interpreted with caution as this technology is currently approved for research use only. The MILLIPLEX^®^/xMAP Human Circulating Cancer Biomarker Magnetic Bead Panel 1 assay has proven to be a stable and accurate method for biomarker detection [41], however, in the case of CA125 a good, but not perfect, correlation of multiplex with routine immunoassay measurements was shown [42].

## 5. Conclusions

Our study demonstrated satisfactory diagnostic performance (c-index around 0.9 in each case) of the two Combined Models 2021, the Proteomic Model 2017, and the CPH-I, with no clear superiority of either model. All models outperformed the ROMA-50. 

The discriminative ability of all three “2021” models in patients aged 50+ was exceptionally good. The opposite was observed for premenopausal patients and patients missed by normal CA125 measurements. Given the good to excellent discriminative ability of our models in patients with suspicious adnexal mass, a further validation in a general postmenopausal population with an assumed low point prevalence (<1%) of OC could clarify the suitability of these models for population screening as well. 

Adding age as a continuous variable to each of the models resulted in a significant improvement in its diagnostic performance. As single parameters, CA125 and OPN demonstrated the best diagnostic parameters, while MIF and PRL the worst ones. 

## Figures and Tables

**Figure 1 cancers-14-01780-f001:**
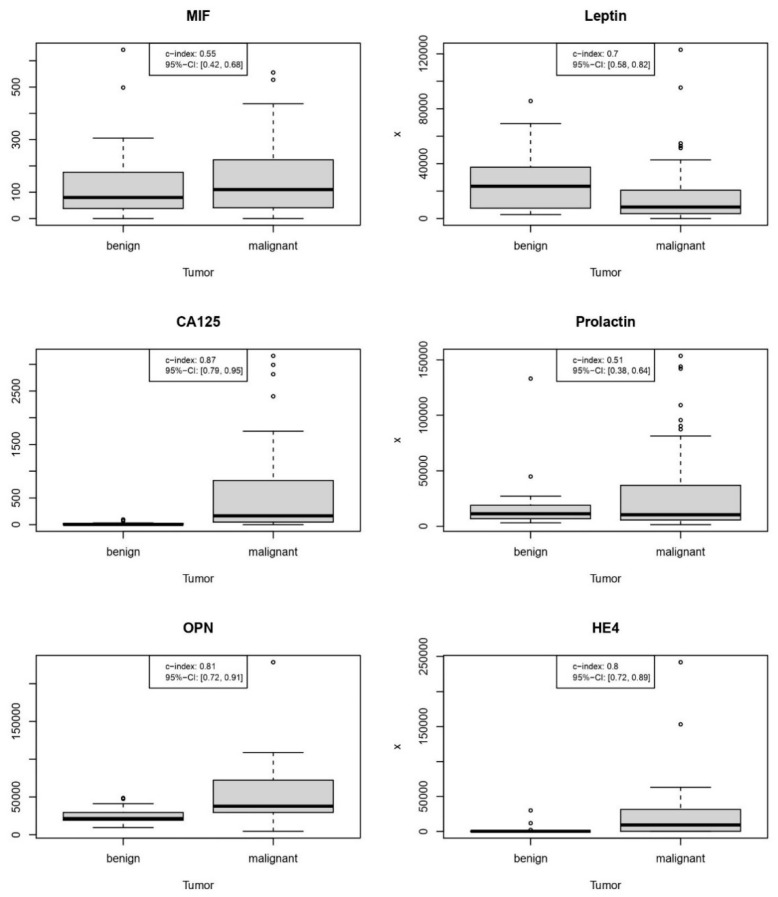
Diagnostic performance of individual protein markers in the validation cohort.

**Figure 2 cancers-14-01780-f002:**
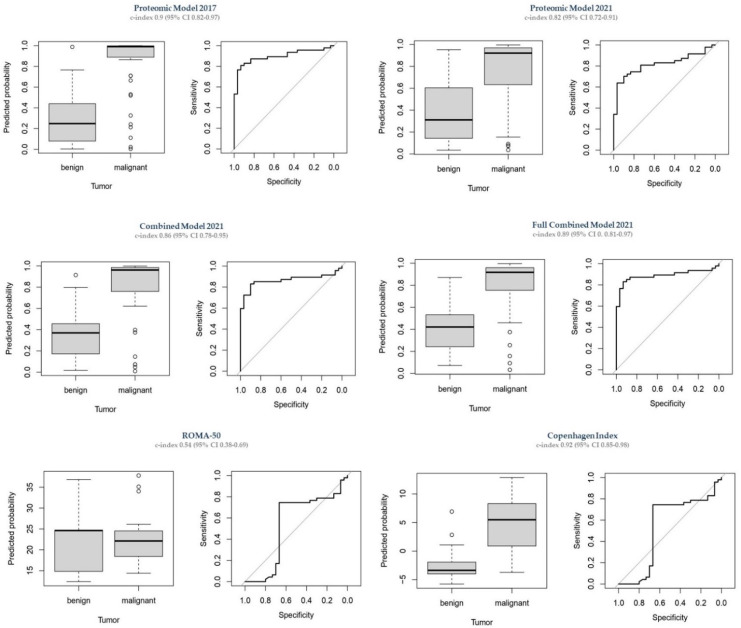
Diagnostic performance of all models in the validation cohort.

**Figure 3 cancers-14-01780-f003:**
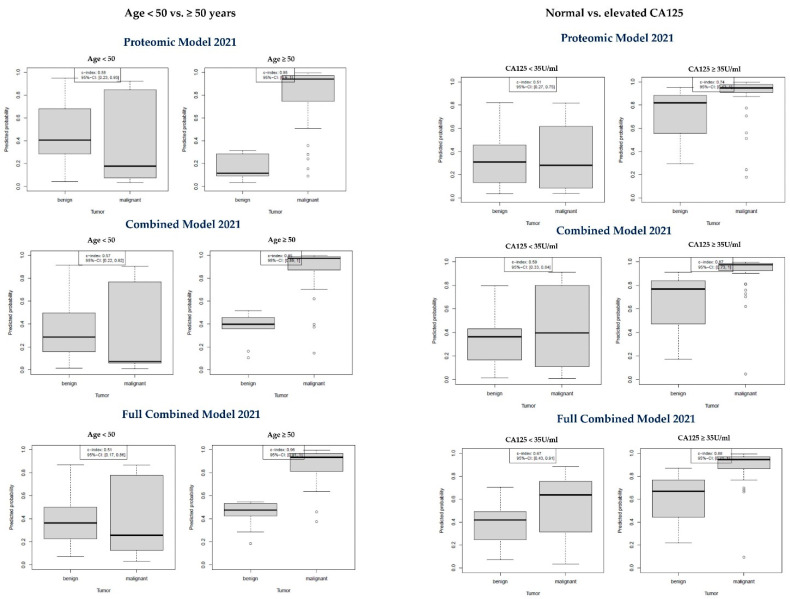
Diagnostic performance of the new models by age (cutoff: 50 years) and CA125 level (cutoff 35 U/mL).

**Table 1 cancers-14-01780-t001:** Characteristics of the training and validation cohorts.

Variable	Training Cohort (*n* = 279)			Validation Cohort (*n* = 77)		
	*n*	Mean or %	SD	*n*	Mean or %	SD
Clinical center						
Berlin	70	25%		0	0%	
Freiburg	72	26%		0	0%	
Innsbruck	0	0%		38	49%	
Leuven	0	0%		39	51%	
Vienna	137	49%		0	0%	
Menopausal status						
<50 years	95	34.1%		27	35.1%	
≥50 years	184	65.9%		50	64.9%	
CA125						
<35 U/mL	116	42%		38	49%	
≥35 U/mL	163	58%		39	51%	
Age (years)	279	55.2	15.6	77	56.3	15.6
CA125 (U/mL)	279	444.8	751.5	77	341.6	690.3
HE4 (pg/mL)	279	28,051	107,318.9	77	15,987.4	35,499
OPN (pg/mL)	279	48,120.4	81,244.2	77	41,656.3	33,348.7
PRL (pg/mL)	279	47,622.9	78,443.3	77	25,950.3	36,547.4
MIF (pg/mL)	279	1788.2	2256.9	77	146.8	147.5
Leptin (pg/mL)	279	19,381.6	24,482.9	77	22,090.3	24,032.8

**Table 2 cancers-14-01780-t002:** Discriminative ability of single markers and models.

Marker or Model	C-Index (95% CI)	Classification Cutoff	Accuracy	Sensitivity	Specificity	PPV	NPV
Age (years)	0.76 (0.64–0.88)	51.46	0.78	0.85	0.67	0.80	0.74
CA125 (U/mL)	0.87 (0.79–0.95)	32.39	0.83	0.79	0.90	0.92	0.73
OPN (pg/mL)	0.81 (0.72–0.91)	24,850.70	0.82	0.89	0.70	0.82	0.81
HE4 (pg/mL)	0.8 (0.72–0.89)	2996.17	0.78	0.68	0.93	0.94	0.65
Leptin (pg/mL)	0.7 (0.58–0.82)	16,428.78	0.70	0.70	0.70	0.79	0.60
MIF (pg/mL)	0.55 (0.42–0.68)	95.17	0.58	0.55	0.63	0.70	0.48
PRL (pg/mL)	0.51 (0.38–0.64)	5065.74	0.51	0.21	0.97	0.91	0.44
Proteomic Model 2017	0.9 (0.82–0.97)	0.69	0.86	0.81	0.93	0.95	0.76
Proteomic Model 2021	0.82 (0.72–0.91)	0.85	0.77	0.64	0.97	0.97	0.63
Combined Model 2021	0.86 (0.78–0.95)	0.69	0.86	0.83	0.90	0.93	0.77
Full Combined Model 2021	0.89 (0.81–0.97)	0.67	0.87	0.83	0.93	0.95	0.78
ROMA-50	0.54 (0.38–0.69)	24.45	0.71	0.74	0.67	0.78	0.62
CPH-I	0.92 (0.85–0.98)	−0.72	0.86	0.83	0.90	0.93	0.77

**Table 3 cancers-14-01780-t003:** Measures of diagnostic accuracy broken down by age (cutoff 50 years).

	Proteomic Model 2021		Combined Model 2021		Full Combined Model 2021	
	Age < 50	Age ≥ 50	Age < 50	Age ≥ 50	Age < 50	Age ≥ 50
Patient number	27	50	27	50	27	50
C-index (95% CI)	0.58 (0.23–0.93)	0.95 (0.9–1.0)	0.57 (0.22–0.92)	0.95 (0.89–1.0)	0.51 (0.17–0.86)	0.96 (0.91–1.0)
Threshold	0.18	0.34	0.09	0.57	0.17	0.59
Overall accuracy	0.81	0.92	0.81	0.94	0.78	0.96
Sensitivity	0.57	0.90	0.57	0.92	0.43	0.95
Specificity	0.90	1.00	0.90	1.00	0.90	1.00
PPV	0.67	1.00	0.67	1.00	0.60	1.00
NPV	0.86	0.71	0.86	0.77	0.82	0.83

**Table 4 cancers-14-01780-t004:** Diagnostic performance broken down by CA125 (cutoff 35 U/mL).

	Proteomic Model 2021		Combined Model 2021		Full Combined Model 2021	
	Normal	Elevated	Normal	Elevated	Normal	Elevated
Patient number	38	39	38	39	38	39
C-index (95% CI)	0.51 (0.27–0.75)	0.74 (0.43–1.0)	0.59 (0.33–0.84)	0.87 (0.73–1.0)	0.67 (0.43–0.91)	0.88 (0.71–1.0)
Threshold	0.09	0.85	0.71	0.92	0.60	0.87
Overall accuracy	0.74	0.82	0.82	0.79	0.82	0.74
Sensitivity	0.36	0.83	0.45	0.78	0.55	0.72
Specificity	0.89	0.67	0.96	1.00	0.93	1.00
PPV	0.57	0.97	0.83	1.00	0.75	1.00
NPV	0.77	0.25	0.81	0.27	0.83	0.23

## Data Availability

The data presented in this study are available on reasonable request from the corresponding author.

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
