# Peer review of "Biomarker-Based Models for Preoperative Assessment of Adnexal Mass: A Multicenter Validation Study"

_cancers, 2022, doi:10.3390/cancers14071780_

Round 1
Reviewer 1 Report
It has been a pleasure reviewing this comprehensive presentation that is very well-written. The context of the presentation is excellent. The novel findings that are presented highlight the applicability chiefly to older women and the limitations of the data set does not allow application to early-stage ovarian cancers. The authors are upfront about discussing more limited applicability to certain ovarian cancer histotypes. In this regard, is it possible there would be a benefit in examining this approach after TVS? (I am aware of their aim to forecast its utility when TVS is not available). An apparent detraction of the presentation is the absence of "n" being included in the figures and tables, which definitely should be added in the revision. Similarly, the results of the training cohort should be used to estimate the power/sample size of the validation/diagnostic cohort and this should be introduced in the revision. I feel that the language of lines 325-326 significantly conveys the positive findings of this presentation and should be exactly reiterated in the abstract (at line 51 or 60?) and considered for the simple summary at line 33.
I recommend adding JAMA Oncol. doi:10.1001/jamaoncol.2021.6987 to reference 1-2. Also due to their related presentation in the MDPI family of publications, the authors should add and discuss the following: https://doi.org/10.3390/diagnostics11081440 https://doi.org/10.3390/diagnostics11020173
After these minor revisions, I am recommending acceptance of this very fine manuscript.
Author Response
Dear Reviewer,
We are grateful for your appreciation and for your friendly and valuable comments!
We have improved our manuscript based on your suggestions.
Here we have responded to your comments one by one.
In the manuscript, newly added sentences or paragraphs are marked in red.
REVIEWER 1 Report
Comments and Suggestions for Authors
It has been a pleasure reviewing this comprehensive presentation that is very well-written. The context of the presentation is excellent. The novel findings that are presented highlight the applicability chiefly to older women and the limitations of the data set does not allow application to early-stage ovarian cancers. The authors are upfront about discussing more limited applicability to certain ovarian cancer histotypes.
We are really grateful for your appreciation!
In this regard, is it possible there would be a benefit in examining this approach after TVS? (I am aware of their aim to forecast its utility when TVS is not available).
We addressed this issue in the Discussion (lines 841-842, see also the new Ref. 64).
An apparent detraction of the presentation is the absence of "n" being included in the figures and tables, which definitely should be added in the revision.
We provided the information about “n” (patient number) in Table 2 and Table 3.
Similarly, the results of the training cohort should be used to estimate the power/sample size of the validation/diagnostic cohort and this should be introduced in the revision.
Thank you for this valuable comment. We agree that sample size calculations based on performance measures in the training cohort should be conducted during the planning phase of an external validation study. Since our study included only an internal validation with retrospective data, we did not do any sample size calculations, which would probably suggest a higher sample size. In newly planned external validation study, aiming at further investigations of the markers and the generalizability of the proteomic models, the calculations are of utmost importance. We added the following text in the limitations to address this issue: “To clarify the generalizability of the Proteomic Models to general screening populations or others, external validation studies need to be conducted since here only internal validation was performed with a retrospective study design and without sample size calculations ” (lines 890-893).
I feel that the language of lines 325-326 significantly conveys the positive findings of this presentation and should be exactly reiterated in the abstract (at line 51 or 60?) and considered for the simple summary at line 33.
We followed this suggestion and inserted the following sentence in the abstract: “Notably, although combining values of only four proteins with age, the Combined Model 2021 performed comparably to the Full Combined Model 2021” (lines 58-60).
I recommend adding JAMA Oncol. doi:10.1001/jamaoncol.2021.6987 to reference 1-2. Also due to their related presentation in the MDPI family of publications, the authors should add and discuss the following:
https://doi.org/10.3390/diagnostics11081440
https://doi.org/10.3390/diagnostics11020173
Thank you for these suggestions. We cited and discussed the recommended publications (lines 830-833 and 841-842; Ref. 3, Ref. 63 and Ref. 64).
After these minor revisions, I am recommending acceptance of this very fine manuscript.
Thank you!
Thank you, too!
Reviewer 2 Report
Very interesting study by Watrowski and colleagues that evaluates new predictive biomarker models in ovarian cancer. The authors note that the validation cohort is rather small and further investigation of the combined model proposed here are certainly warranted based on the potential performance of the model. Otherwise, the statistical assessment of the data is appropriate and the model shows much promise. I do have a couple of questions that would be useful to discuss to further elaborate on the model.
Major Points:
- Was obesity, family history, HRT or other known risk factors considered for the combined model? There is no mention of why these would have been excluded while age was included. These are generally important and routine to consider during patient assessment.
- While appropriate for indicating surgical consult, early detection remains one of the most important areas to improve in the treatment of ovarian cancer. Were there any differences in correctly identifying early stage (I/II) versus advanced (IIb/IV) disease regarding the plasma levels of OPN and the other proteins selected for this model, either as single variables or in multivariate analysis?
Author Response
Dear Reviewer,
Thank you very much for your valuable comments! We followed your recommendations and improved our manuscript. Here we have responded to your remarks one by one. In the manuscript, newly added sentences or paragraphs are marked in red.
REVIEWER 2 Report
Comments and Suggestions for Authors
Very interesting study by Watrowski and colleagues that evaluates new predictive biomarker models in ovarian cancer.
Thank you for your appreciation!
The authors note that the validation cohort is rather small and further investigation of the combined model proposed here are certainly warranted based on the potential performance of the model. Otherwise, the statistical assessment of the data is appropriate and the model shows much promise. I do have a couple of questions that would be useful to discuss to further elaborate on the model.
Major Points:
Was obesity, family history, HRT or other known risk factors considered for the combined model?
Thank you for this valuable remark. Indeed, obesity or family history are relevant risk factors which can be part of more complex diagnostic models, e.g. nomograms. We reflected the role of obesity (line 841-842) and supplied this information with a new reference (Anastasi et al. 2016, Ref. 65).
There is no mention of why these would have been excluded while age was included. These are generally important and routine to consider during patient assessment.
As our intention was to develop biomarker-based models utilizing only a minimal blood sample, we did not include anamnestic variables in our models. The only exception was age, which is always and obligatorily available (in order to identify the blood sample) and is a variable with an eminent biological role.
While appropriate for indicating surgical consult, early detection remains one of the most important areas to improve in the treatment of ovarian cancer. Were there any differences in correctly identifying early stage (I/II) versus advanced (IIb/IV) disease regarding the plasma levels of OPN and the other proteins selected for this model, either as single variables or in multivariate analysis?
Unfortunately, as mentioned in “Results” (line 407-409) and “Discussion” (line 864-868), the proportion of early stages in our dataset (16% and 5% in the training and validation cohorts, respectively) is one of the limitations of our study. We apologize that we could not assess the usefulness of our models for the early OC diagnosis.